# An Ultrasonic Laminated Transducer for Viscoelastic Media Detection

**DOI:** 10.3390/s21217188

**Published:** 2021-10-29

**Authors:** Shunmin Yang, Wenai Song, Yifang Chen, Lu Yang, Mingquan Wang, Yongjian Lian, Kangchi Liu

**Affiliations:** 1Science and Technology on Electronic Test and Measurement Laboratory, North University of China, Taiyuan 030051, China; songwenai@nuc.edu.cn (W.S.); tyyanglu@nuc.edu.cn (L.Y.); wangmq@nuc.edu.cn (M.W.); lyj@nuc.edu.cn (Y.L.); b1705013@st.nuc.edu.cn (K.L.); 2Department of Mechanical Engineering, Tsinghua University, Beijing 100084, China; chenyifang@mail.tsinghua.edu.cn

**Keywords:** ultrasonic laminated transducer, viscoelastic media, resonance frequency, amplitude gain, ultrasonic feature scanning system

## Abstract

Based on the principle of underwater transducers, an ultrasonic four-laminated transducer with a frequency of 1 MHz was proposed to solve the problem of large energy attenuation when ultrasonic waves propagate in viscoelastic media. First, this study targeted solid rocket propellant as the research object, and the energy attenuation characteristics of ultrasonic waves propagating in viscoelastic media were analyzed through the derivation of the wave equation. Second, the structure of a four-laminated transducer with a frequency of 1 MHz was designed, and the resonance frequency was obtained by a graphical method. The sound field simulation and experimental results showed that the gain of the four-laminated transducer was 15 dB higher than that of the single-wafer transducer. An ultrasonic feature scanning system was built to complete the qualitative and quantitative detection of the smallest artificial hole (ϕ2 mm × 10 mm). Finally, two different natural defects were scanned, and the results were compared with those obtained using an industrial computed tomography detection system. The results showed that the ultrasonic method was more accurate in characterizing two natural defects. The primary cause was that the industrial CT was not sensitive to defects parallel to the incident direction of the ray. Therefore, this study not only achieved the qualitative and quantitative nondestructive testing of solid rocket propellants, but also provides an important reference for other viscoelastic components.

## 1. Introduction

### 1.1. Purpose and Significance

For elastic media, the material stores energy without dissipation during deformation, but for viscoelastic media, the material dissipates a large amount of energy during deformation. Therefore, the deformation process of viscoelastic materials with time can be summarized as follows [1].

Creep: Under constant loading, the deformation will gradually increase.Relaxation: Under constant strain, the stress will gradually weaken.Hysteresis: The strain response of the material lags behind the stress, causing the stress–strain curve during a loading process to form a hysteresis loop. The area under the hysteresis loop represents the energy loss during loading and unloading.Strain sensitivity: Some physical quantities that reflect the mechanical properties of materials, such as the Young’s modulus, shear modulus, and Poisson’s ratio, are generally related to the strain rate (or time).

Viscoelastic materials can be described by a “spectrum”. At the far left end of this spectrum are elastic media, and at the far right end are classical viscous fluids. Many actual materials exhibit mechanical properties between the two extremes of elasticity and viscosity. The viscoelastic properties can be represented as a combination of the elastic and viscous properties in a certain proportion. In general, solid polymers (such as nylon and plastics) and metals are close to the elastic end, while viscoelastic fluids (such as polymer solutions) are close to the viscous end, and the properties of molten polymer materials are between the two. Where any specific material is located on the “spectrum” of viscoelastic materials depends not only on the conditions of itself but also on the working conditions, such as the temperature and loading rate. Steel is a solid under normal conditions, but it is no different from a fluid under high-speed impacts. Silly Putty is highly malleable under normal circumstances, but it can bounce back like a ball when it hits the ground quickly. Therefore, research on the viscoelastic properties of materials or components has important value and practical significance.

Under normal conditions, many materials have the characteristics of elastic solids and viscous fluids at the same time, such as industrial materials (including plastics, rubbers, solid rocket propellants, resins, paints, glasses, ceramics, and metals), geological materials (including rocks, soil, asphalt, concrete, petroleum, and minerals), biological materials (including muscles, tendons, bones, and blood), and raw materials in light and commodity industries (including textile fiber, pulp, cosmetics, oil, and food). The application of viscoelastic materials involves the fields of machinery, construction, transportation, information, and aerospace as well as the chemical industry. Determining how to evaluate or monitor the safety of viscoelastic materials during service is an important issue related to economic and social development. This study targeted solid rocket propellants as the research object to explore nondestructive evaluation methods for viscoelastic media.

The non-destructive testing of solid propellants is crucial for ensuring the safety and reliability of rocket launch operations. Solid rocket propellants, which are granular heterogeneous composite materials, are formed mainly from a polymer compound as the matrix, along with solid oxidants and other particles that serve as the core components of rocket engines [2]. However, defects will be introduced during production, storage, and service, such as discontinuities, cracks, pores, and inclusions, which cause the burning area of the propellant to deviate from the designed burning surface. These deviations can result in highly unpredictable consequences, including some extent of misaligned thrust resulting in trajectory errors to rocket explosions during flight. Therefore, the nondestructive testing of solid propellants is important to ensure the safety and reliability of rocket launch operations. At present, a number of nondestructive testing methods have been applied for detecting defects in solid propellants, such as ultrasound [3], X-ray [4], microwave [5], infrared [6], and industrial computed tomography (CT) [7] methods. Among these methods, ultrasound and, in particular, CT imaging combine a high detection sensitivity with the ability to accurately locate defects in three-dimensional space. In particular, ultrasonic testing (UT) has been more widely used owing to its many advantages, such as its low design and maintenance costs, as well as a complete absence of potentially harmful radiation [8]. Therefore, ultrasonic testing is the best choice among many nondestructive testing methods for solid rocket propellants.

### 1.2. Viscoelastic Properties

A solid rocket propellant is mainly composed of a high-energy combustion agent, binder, oxidant, plasticizer, and curing agent, and its mechanical properties are viscoelastic [9]. In recent years, many scholars have performed considerable work on the acoustic properties of viscoelastic media. Imperial et al. [10] performed a numerical simulation on the propagation of transient waves in anisotropic viscoelastic layered materials. By analyzing the Kelvin–Voight, Maxwell, and Zener models, the mortar element method was used to merge the effective transfer conditions between the layers to solve the attenuation in the anisotropic layer. Larcher et al. [11] used the ultrasonic measurement method to realize the viscoelastic mechanical characterization of asphalt materials. Through P-wave transmission testing, the wave velocity and attenuation factor were obtained, and then the high-frequency complex modulus was calculated. A transient model of guided waves in a viscoelastic cylinder was established by Bause et al. [12], and the scale boundary finite element method was applied to calculate the phase velocity dispersion effectively. Aksoy [13] proposed a method based on the combination of acoustic wave transmission and broadband spectroscopy to realize the measurement of the phase velocity and attenuation coefficient of ultrasonic waves. Shih et al. [14] combined the Lamb wave model with an ultrasonic micro-elastic imaging system to obtain the accurate viscoelastic properties of thin tissue, and acquired the phase velocity and attenuation coefficient of viscoelastic materials from k-space by a pulse method. By measuring the the reflection coefficient, wave velocity and attenuation coefficient, Lin et al. [15] calculated the storage moduli, loss moduli, and loss angles of viscoelastic materials with an air-coupled ultrasonic method. Li et al. [16] evaluated the viscoelasticity of four rubber materials with the ultrasonic surface reflection method and optimized the actual performance by changing the size of the transducer diameter. Lu et al. [17] deduced the high-order finite difference equation for the Kelvin–Voight model and simulated the propagation process of ultrasonic waves in several viscoelastic concrete media. Hutchins et al. [18] studied the multiple scattering of sound waves in a solid viscoelastic media wrapped by a cylindrical steel bar. The experimental results showed that the combination of a piezoelectric composite transducer and pulse compression process could effectively improve the signal-to-noise ratio of the strong attenuation signal in viscoelastic media. Li et al. [19] studied the propagation of ultrasonic guided waves in viscoelastic isotropic materials and compared the differences in scattering and attenuation between the thin asphalt on a steel substrate and single asphalt. The research results showed that the ultrasonic wave exhibited large attenuation when propagating in viscoelastic media, which introduces a huge challenge for detecting solid rocket propellants. Based on the principle of underwater transducers, an ultrasonic four-laminated transducer with a frequency of 1 MHz was proposed to solve the problem of large energy attenuation.

### 1.3. Piezoelectric Laminated Transducer

In the research on ultrasonic laminated transducers, many scholars have carried out considerable work. Based on an accurate analytical model of piezoelectric laminated transducers with variable frequencies, the multi-frequency characteristics of laminated transducers can be realized by adjusting the resistance and the number of layers of active components [20]. Several cascaded laminated transducers were developed and the performance could be optimized by reasonably selecting the position of the piezoelectric wafers [21]. Gamboa et al. [22] evaluated the mechanisms and performances of various transducers laminated with lead zirconate titanate (PZT) for effective electromechanical energy conversion, and the power density of the 1:3 composite piezoelectric transducer was the highest. Li et al. [23] designed a new type of broadband piezoelectric transducer and wafers with different resonance frequencies could increase the transducer bandwidth and the transmitting/receiving power. Wei et al. [24] proposed the PSpice lossy model of sandwich piezoelectric ultrasonic transducers and obtained the resonance and anti-resonance frequencies according to the model. The research results showed that the best excitation frequency of the pitch–catch setup was not necessarily the resonance frequency of the ultrasonic transducer because the resonance frequency was obtained under no loading. Piao et al. [25] studied the effect of the thickness of the piezoelectric wafer on the performances of ultrasonic sandwich transducers. The natural frequency of the radial mode of the transducer was not affected by the stacking and thickness of the piezoelectric wafer. However, it had a greater impact on the thickness mode. Deng et al. [26] proposed an electrical method that connected the piezoelectric wafers on the upper and lower transducers with resistors in series. This method could reduce the transmitting voltage response (TVR) fluctuations of the dual-excitation piezoelectric transducer in the working frequency band from 12 to 7.3 dB. Lin [27] analyzed the impact of parallel matching inductance on piezoelectric transducers. When the parallel matching inductance increased, the anti-resonance frequency decreased, and the resonance frequency did not change. Due to the effects of the size, loss, and loading resistance through the electromechanical equivalent circuit of cascaded piezoelectric transducers, when the distance between the stacked wafers of the transducer increased, the resonance/anti-resonance frequency and the effective electromechanical coupling coefficient had the maximum value. The mechanical quality factor had a maximum value and a minimum value, and the electroacoustic efficiency was reduced [28]. For an accurate theoretical model of the piezoelectric stack actuator, ignoring the electrode and protective layer will greatly affect the accuracy of the prediction model [29]. An ultrasonic energy transfer (UPT) system was built, which used piezoelectric stacks in series and parallel configurations. Based on the output voltage and power characteristics of the system, the power transmission efficiency on a 1.0 mm steel plate could be increased by 26% [30]. A new type of piezoelectric stack intelligent integrated transducer structure (PiSSA) was proposed, which used piezoelectric wafers connected in series or in parallel. Compared with the traditional single wafer, the efficiencies and output performances of the wafers were higher [31]. Meng et al. [32] used the equivalent elastic method to obtain the radial and longitudinal electromechanical equivalent circuits of a sandwich transducer and derived the resonance frequency equation. The thickness or radius of the piezoelectric wafer had a significant influence on the longitudinal or radial resonance frequency. Chen et al. [33] derived the equations of the resonance frequency and amplification factor of the quarter-wave cone transducer based on electromechanical equivalent circuit theory. The addition of pre-stressed bolts could increase the resonant frequency of the stacked transducer.

Two conclusions can be drawn from the literature review. One is that the energy of the laminated transducer is higher than that of the single-wafer transducer, which can account for the strong attenuation of ultrasonic waves propagating in viscoelastic media. The other is that the laminated technology of piezoelectric wafers is mainly used in underwater transducers, and no relevant applications have been found for the ultrasonic testing of industrial components. Due to the different structures, the underwater acoustic transducer cannot be directly used for industrial component detection. Therefore, the ultrasonic laminated transducer proposed in this paper has important application value and significance. The differences between the ultrasonic laminated transducer with a frequency of 1 MHz proposed in this paper and the existing underwater transducers are shown in Table 1.

### 1.4. Contributions of Present Work

In this study of ultrasonic laminated transducers for detecting viscoelastic media, the following tasks were completed.

Based on the correspondence principle of simple harmonic waves, the wave equation in viscoelastic media was derived. Since the equation is a function of the angular frequency ω, ultrasonic waves propagate in viscoelastic media with greater attenuation than that in elastic media.According to the amplitude–frequency relationship in viscoelastic media, by setting the independent variable range of 950–1050 kHz for the transcendental equation, the 1-MHz resonance frequency of the laminated transducer was obtained by a graphical method.The sound field simulation and experimental results showed that the gain of the four-laminated transducer was 15 dB higher than that of the single-wafer transducer.An ultrasonic feature scanning system was built to realize the qualitative and quantitative detection of the smallest artificial hole (⌀2 mm × 10 mm).Two different natural defects were scanned, and the results were compared with those obtained using an industrial CT detection system. The results showed that the ultrasonic method was more accurate in the characterization of the two natural defects. The primary reason was that the industrial CT system was not sensitive to defects parallel to the incident direction of the ray.

## 2. Principle of Laminated Transducer

### 2.1. Correspondence Principle of Simple Harmonic Wave

Under the condition of a small deformation, a viscoelastic strain displacement, three-dimensional constitutive and dynamic equations are as follows [34]:(1)$$\begin{array}{c} \epsilon_{ij}\left(x_{i},t\right)=\frac{1}{2}\left[u_{i,j}\left(x_{i},t\right)+u_{j,i}\left(x_{i},t\right)\right],\\ \sigma_{ij}\left(x_{i},t\right)=\delta_{ij}\lambda \left(t\right)\cdot d\epsilon_{kk}+2\mu \left(t\right)\cdot d\epsilon_{ij},\\ \sigma_{ij,j}\left(x_{i},t\right)+f_{i}=\rho \frac{\partial^{2}u_{i}\left(x_{i},t\right)}{\partial t^{2}}, \end{array}$$
where $\delta_{ij}$ is the Kronecker symbol, i,j,k represent the *x*-, *y*-, and *z*-directions, respectively, $f_{i}$ is the component of the volumetric force, λ(t) and μ(t) are viscoelastic material functions, and $\mu_{i}=\mu_{i}\left(x_{i},t\right)$ is the displacement component.

From Equation (Equation 1), the three-dimensional equation of motion of viscoelastic media can be deduced as follows:(2)$$\left[\lambda \left(t\right)+\mu \left(t\right)\right]*du_{j,ji}+\mu \left(t\right)*du_{i,jj}+f_{i}=\rho \frac{\partial^{2}u_{i}}{\partial t^{2}},\left(i,j=1,2,3\right),$$
where the symbol “*” is a temporal convolution product, except when stated otherwise. The particle velocity is $v_{i}\left(x_{i},t\right)=\frac{\partial u_{i}}{\partial t}$ and the physical forces are neglected. $\theta =u_{j,j}$, $\theta_{,i}=u_{j,ji}$, and $\nabla^{2}u_{i}=u_{i,jj}$ are substituted into Equation (Equation 2), yielding the following:(3)$$\left[\lambda \left(t\right)+\mu \left(t\right)\right]*d\theta_{,i}+\mu \left(t\right)*d\nabla^{2}u_{i}=\rho \frac{\partial v_{i}}{\partial t},$$

That is,
(4)$$\begin{array}{c} \rho \frac{\partial v_{x}}{\partial t}=\left[\lambda \left(t\right)+\mu \left(t\right)\right]*d\left(\frac{\partial \theta }{\partial x}\right)+\mu \left(t\right)*d\nabla^{2}u_{x},\\ \rho \frac{\partial v_{y}}{\partial t}=\left[\lambda \left(t\right)+\mu \left(t\right)\right]*d\left(\frac{\partial \theta }{\partial y}\right)+\mu \left(t\right)*d\nabla^{2}u_{y},\\ \rho \frac{\partial v_{z}}{\partial t}=\left[\lambda \left(t\right)+\mu \left(t\right)\right]*d\left(\frac{\partial \theta }{\partial z}\right)+\mu \left(t\right)*d\nabla^{2}u_{z}, \end{array}$$
where $\left(v_{x},v_{y},v_{z}\right)$ are the three components of the velocity vector, and $\left(u_{x},u_{y},u_{z}\right)$ are the three components of the displacement vector.

Under the condition of a small deformation, Equations (Equation 1)–(Equation 4) are the basic equations of viscoelastic media. Here, we only discuss the fluctuations of the frequency ω of the displacement $u_{i}\left(x,t\right)$ with time *t*, namely,
(5)$$u_{i}\left(x_{i},t\right)={\tilde{u}}_{i}e^{i\omega t},$$
where $\tilde{u}\left(x_{i}\right)$ is only a function of the coordinate $x_{i}$, which has nothing to do with *t* and is generally complex. The conditions for this movement are as follows. The boundary conditions (boundary force and boundary displacement) and volume force all changed with the same angular frequency ω over time *t*. In the same way, all the strain and stress components also made simple harmonic changes with an angular frequency ω, namely,
(6)$$\epsilon_{ij}\left(x_{i},t\right)={\tilde{\epsilon }}_{ij}e^{i\omega t},$$
(7)$$\sigma_{ij}\left(x_{i},t\right)={\tilde{\sigma }}_{ij}e^{i\omega t}.$$

Equations (Equation 5)–(Equation 7) are substituted into Equation (Equation 1), and the governing equation of the simple harmonic wave in a linear viscoelastic media, which can be represented as a displacement, is solved as follows:(8)$$\left[\lambda^{*}\left(i\omega \right)+\mu^{*}\left(i\omega \right)\right]\tilde{\theta_{,i}}+\mu^{*}\left(i\omega \right){▽}^{2}\tilde{u_{i}}+{\tilde{f}}_{i}+\rho \omega^{2}{\tilde{u}}_{i}=0,$$
where $\mu^{*}\left(i\omega \right)$ is the complex shear modulus of the viscoelastic media, $\lambda^{*}\left(i\omega \right)=K^{*}\left(i\omega \right)-\frac{2}{3}\mu^{*}\left(i\omega \right)$, $K^{*}\left(i\omega \right)$ is the complex bulk modulus, $\tilde{\theta }\left(x_{i}\right)={\tilde{u}}_{j,j}$ and $\tilde{\theta_{,i}}\left(x_{i}\right)={\tilde{u}}_{j,ji}$, which are only functions of $x_{i}$ and have nothing to do with *t*, and $f_{i}\left(x_{i},t\right)=\tilde{f}\left(x_{i}\right)e^{i\omega t}$, where ${\tilde{f}}_{i}\left(x_{i}\right)$ is related to $x_{i}$. The governing equation of the simple harmonics in an elastic media is as follows [34]:(9)$$\left(\lambda +\mu \right)\theta_{,i}+\mu {▽}^{2}u_{i}+f_{i}+\rho \omega^{2}u_{i}=0.$$

By comparing and analyzing Equations (Equation 8) and (Equation 9), the correspondence between the elastic and viscoelastic media was obtained, as shown in Table 2. This is the correspondence principle of a simple harmonic wave.

### 2.2. Wave Equation in Elastic Media

For a homogeneous, isotropic, and infinite elastic medium, we assume that the velocity of any plane wave is $c_{0}$. In general, the plane wave propagates along the *x*-direction, and the displacements $u_{x}$, $u_{y}$, and $u_{z}$ are functions of $\xi =x-c_{0}t$, i.e.,
(10)$$\begin{array}{c} u_{x}=u_{x}\left(x-c_{0}t\right),\\ u_{y}=u_{y}\left(x-c_{0}t\right),\\ u_{z}=u_{z}\left(x-c_{0}t\right). \end{array}$$

Substituting Equation (Equation 10) into Equation (Equation 4) yields the following:(11)$$\begin{array}{c} \rho c_{0}^{2}\frac{\partial^{2}u_{x}}{\partial \xi^{2}}=\left(\lambda +2\mu \right)\frac{\partial^{2}u_{x}}{\partial \xi^{2}},\\ \rho c_{0}^{2}\frac{\partial^{2}u_{y}}{\partial \xi^{2}}=\mu \frac{\partial^{2}u_{y}}{\partial \xi^{2}},\\ \rho c_{0}^{2}\frac{\partial^{2}u_{z}}{\partial \xi^{2}}=\mu \frac{\partial^{2}u_{z}}{\partial \xi^{2}}. \end{array}$$

Equation (Equation 11) has only two possible solutions if $\frac{\partial^{2}u_{x}}{\partial \xi^{2}}$, $\frac{\partial^{2}u_{y}}{\partial \xi^{2}}$, and $\frac{\partial^{2}u_{x}}{\partial \xi^{2}}$ are not simultaneously zero. One solution is for the longitudinal wave, and it is:(12)$$\begin{array}{c} c_{0}^{2}=\frac{\lambda +2\mu }{\rho }=c_{L}^{2},\\ \frac{\partial^{2}u_{y}}{\partial \xi^{2}}=\frac{\partial^{2}u_{z}}{\partial \xi^{2}}=0. \end{array}$$

In this case, there is only an *x*-axis disturbance and the displacement solution is:(13)$$\begin{array}{c} u_{y}=u_{z}=0,\\ u_{x}\left(x,t\right)=\tilde{u_{x}}\left(x\right)e^{i\omega t},\\ \tilde{u_{x}}=Aexp\left(-i\omega x\sqrt{\frac{\rho }{\lambda +2\mu }}\right). \end{array}$$

The other solution is for a transverse wave, and it is:(14)$$\begin{array}{c} c_{0}^{2}=\frac{\mu }{\rho }=c_{T}^{2},\\ \frac{\partial^{2}u_{x}}{\partial \xi^{2}}=0. \end{array}$$

In this case, there is only a *y*- or *z*-axis disturbance, and the displacement solution is:(15)$$\begin{array}{c} u_{x}=u_{z}=0,\\ u_{y}=\tilde{u_{x}}\left(x\right)e^{i\omega t},\\ \tilde{u_{y}}=Bexp\left(-i\omega x\sqrt{\frac{\rho }{\mu }}\right), \end{array}$$
or
(16)$$\begin{array}{c} u_{x}=u_{y}=0,\\ u_{z}=\tilde{u_{z}}\left(x\right)e^{i\omega t},\\ \tilde{u_{z}}=Cexp\left(-i\omega x\sqrt{\frac{\rho }{\mu }}\right). \end{array}$$

For a homogeneous, isotropic, and infinite elastic medium, the results show that the propagation mode of any plane wave is either a longitudinal wave ($c_{L}$) or a transverse wave ($c_{T}$).

### 2.3. Wave Equation in Viscoelastic Media

According to the correspondence principle of simple harmonic waves, the solution of plane simple harmonics in infinite viscoelastic media can be obtained from the corresponding solution in elastic media [34]. In other words, replacing the constant of the elastic medium in Equation (Equation 13) with the complex function of the viscoelastic medium, the longitudinal wave solution is:(17)$$\begin{array}{c} u_{y}=u_{z}=0,\\ u_{x}\left(x,t\right)=\tilde{u_{x}}\left(x\right)e^{i\omega t},\\ \tilde{u_{x}}=Aexp\left(-i\omega x\sqrt{\frac{\rho }{\lambda^{*}\left(i\omega \right)+2\mu^{*}\left(i\omega \right)}}\right). \end{array}$$

Similarly, the transverse wave solution is:(18)$$\begin{array}{c} u_{x}=u_{z}=0,\\ u_{y}\left(x,t\right)=\tilde{u_{y}}\left(x\right)e^{i\omega t},\\ \tilde{u_{y}}=Bexp\left(-i\omega x\sqrt{\frac{\rho }{\mu^{*}\left(i\omega \right)}}\right), \end{array}$$
or
(19)$$\begin{array}{c} u_{x}=u_{y}=0,\\ u_{z}\left(x,t\right)=\tilde{u_{z}}\left(x\right)e^{i\omega t},\\ \tilde{u_{z}}=Cexp\left(-i\omega x\sqrt{\frac{\rho }{\mu^{*}\left(i\omega \right)}}\right). \end{array}$$

For a solution in viscoelastic media, although only the parameters λ and μ of the solution for elastic media are replaced by $\lambda^{*}\left(i\omega \right)$ and $\mu^{*}\left(i\omega \right)$, the propagation characteristics of ultrasonic waves are significantly different. We obtain the following:The velocities of longitudinal and transverse waves in viscoelastic media are denoted as $c_{Lv}$ and $c_{Tv}$, respectively. We obtain:
(20)$$\begin{array}{c} c_{Lv}=Re\sqrt{\frac{\lambda^{*}\left(i\omega \right)+2\mu^{*}\left(i\omega \right)}{\rho }},\\ c_{Tv}=Re\sqrt{\frac{\mu^{*}\left(i\omega \right)}{\rho }}, \end{array}$$
where $\lambda^{*}\left(i\omega \right)$, $\mu^{*}\left(i\omega \right)$, $c_{Lv}$, and $c_{Tv}$ are functions of the angular frequency ω. Therefore, when ultrasonic waves propagate in viscoelastic media, frequency dispersion will occur.In an ideal elastic medium, the plane wave is not attenuated, while in a viscoelastic medium, the plane wave attenuates as the propagation distance increases. The attenuation coefficients of the longitudinal and transverse wave are denoted as $\alpha_{L}$ and $\alpha_{T}$, respectively, and we have:
(21)$$\begin{array}{c} \alpha_{L}=-\omega Im\sqrt{\frac{\rho }{\lambda^{*}\left(i\omega \right)+2\mu^{*}\left(i\omega \right)}},\\ \alpha_{T}=-\omega Im\sqrt{\frac{\rho }{\mu^{*}\left(i\omega \right)}}, \end{array}$$
where Re represents the real part of the complex number, and Im represents the imaginary part of the complex number.

The ultrasonic attenuation in viscoelastic media is greater than that in elastic media. The primary cause is that the solutions of the wave equation in viscoelastic media are functions of the angular frequency ω. To obtain the same detection resolution as that in elastic media, a four-laminated ultrasonic transducer with a frequency of 1 MHz was proposed.

### 2.4. Structure of Laminated Transducer

The structure of the four-laminated transducer is presented in Figure 1. It is composed of a case, epoxy potting, backing material, piezoelectric wafers, a matching layer, a copper sheet, a ground wire, a signal wire, and a connector. The case protects the internal components of the transducer. The epoxy potting acts as insulation between the case and the internal components. The backing material not only controls the vibration of the piezoelectric wafers but also absorbs the sound waves emitted by the piezoelectric wafers backward as much as possible. With a thickness of half the wavelength, the piezoelectric wafers convert from electrical energy to acoustic energy and vice versa. The matching layer is used to ease the impedance matching between the piezoelectric wafers and the detected object, and the thickness is a quarter of the wavelength. The copper sheet is the common electrode of the piezoelectric wafers connected in series. To transmit electrical signals for the ground and signal wire, the connector is used to connect the signal cable. Due to the superposition of the thick vibrations of the piezoelectric wafers, the four laminated transducer produces higher energy.

### 2.5. Resonance Frequency

In cylindrical coordinates, the equation of motion of a single piezoelectric wafer is:(22)$$\rho \frac{\partial^{2}\xi_{r}}{\partial t^{2}}=\frac{\partial T_{r}}{\partial r}+\frac{T_{r}-T_{\theta }}{r},$$
where ρ is the density, $\xi_{r}$ is the radial displacement component, *t* is the thickness, and $T_{r}$ and $T_{\theta }$ represent the normal stress along the radial and tangential directions, respectively.

Since the electric field is added to the *z*-axis and the boundary effect of the electric field is ignored, only $E_{3}\neq 0$ ($E_{3}$ represents the electric field strength along the *z*-direction), and the piezoelectric equation is:(23)$$\begin{array}{c} S_{r}=s_{11}^{E}T_{r}+s_{12}^{E}T_{\theta }+d_{31}E_{z},\\ S_{\theta }=s_{12}^{E}T_{r}+s_{11}^{E}T_{\theta }+d_{31}E_{z},\\ D_{z}=d_{31}T_{r}+d_{31}T_{\theta }+\varepsilon_{33}^{T}E_{z}, \end{array}$$
where $S_{r}$ and $S_{\theta }$ represent the normal strains along the *r* and θ directions, respectively, $D_{z}$ is the electric displacement along the *z*-direction, $s_{11}^{E}$ and $s_{12}^{E}$ are the piezoelectric constants, $E_{z}$ is the electric field strength along the *z*-direction, $\varepsilon_{33}^{T}$ is the dielectric constant component, and $d_{31}$ is the piezoelectric strain constant component. We obtain:(24)$$\begin{array}{c} T_{r}=\left(\frac{Y_{0}^{E}}{1-\sigma^{2}}\right)\left(S_{r}+\sigma S_{\theta }\right)-\frac{d_{31}Y_{0}^{E}}{1-\sigma }E_{z},\\ T_{\theta }=\left(\frac{Y_{0}^{E}}{1-\sigma^{2}}\right)\left(S_{\theta }+\sigma S_{r}\right)-\frac{d_{31}Y_{0}^{E}}{1-\sigma }E_{z},\\ D_{z}=d_{31}\left(T_{r}+T_{\theta }\right)+\varepsilon_{33}^{T}E_{z}, \end{array}$$
where $Y_{0}^{E}=1/s_{11}^{E}$ is the Young’s modulus, and $\sigma =-s_{12}^{E}/s_{11}^{E}$ is the Poisson’s ratio.

In the same way, the mechanical vibration equation can be obtained:(25)$$F=j\rho vS\left[-\frac{J_{0}\left(ka\right)}{J_{1}\left(ka\right)}+\frac{1-\sigma }{ka}\right]\dot{\xi_{a}}+nV,$$
where *F* is the applied stress, ρ is the density, $v=\sqrt{\frac{Y_{0}^{E}}{\rho (1-\sigma^{2})}}$ is the wave velocity, *S* is the cross-sectional area, *a* is the radius, k=ω/v is the wave number, ω=2πf is the angular frequency, $J_{0}\left(ka\right)$ is the zeroth-order Bessel function of the first kind, $J_{1}\left(ka\right)$ is the first-order Bessel function of first kind, $\dot{\xi_{a}}$ is the resonant vibration speed, $n=\frac{2\pi ad_{31}Y_{0}^{E}}{1-\sigma }$ is the electromechanical conversion factor, and *V* is the applied voltage.

Similarly, the circuit state equation is:(26)$$I=j\omega C_{0}V-n\dot{\xi_{a}},$$
where $C_{0}=\pi a^{2}{\bar{\varepsilon }}_{33}/t$ is the cut-off capacitance, and *I* is the current.

From Equations (Equation 25) and (Equation 26), the electromechanical equivalent diagram of the piezoelectric wafer can be obtained, as shown in Figure 2.

If the piezoelectric wafer vibrates freely, that is, F=0, the resonance frequency equation of a single piezoelectric wafer can be obtained as follows:(27)$$kaJ_{0}\left(ka\right)=\left(1-\sigma \right)J_{1}\left(ka\right).$$

Therefore, the admittance equation of the piezoelectric wafer is:(28)$$Y=j\omega C_{0}+\frac{n^{2}}{j\rho vS[\frac{1-\sigma }{ka}-\frac{J_{0}\left(ka\right)}{J_{1}\left(ka\right)}]}.$$

The laminated transducer is composed of four piezoelectric wafers, and the piezoelectric wafers are connected in parallel on the circuit and in series mechanically. From the circuit point, it is to cascade the same four-terminal network with each other. According to cascade theory in a circuit, the electromechanical equivalent circuit diagram of the laminated transducer can be obtained [35], as shown in Figure 3.

The admittance of the laminated transducer is the superposition of the admittance of each piezoelectric wafer. Thus, the admittance equation of the four-laminated transducers can be deduced as follows:(29)$$Y=4j\omega C_{0}+\frac{n^{2}}{j\rho vS[\frac{1-\sigma }{ka}-\frac{J_{0}\left(ka\right)}{J_{1}\left(ka\right)}]}.$$
when Y→∞, the transducer enters the resonance state, and the vibration frequency at this time is the resonance frequency of the transducer.

In the actual excitation of horizontal shear (SH) waves, the energy E(f) radiated by the transducer is diffused in the frequency domain, and it is assumed that the energy distribution is Gaussian. Therefore, for any excitation frequency $f_{0}$, there is a frequency diffusion interval $\left(f_{0}-3\sigma ,f_{0}+3\sigma \right)$, where σ is the standard deviation. In addition, to model a practical application, the amplitude of any excitation frequency $f_{0}$ must be expressed in the following form [36]:(30)$$\bar{u}\left(\xi ,\zeta ,f_{0}\right)=\int_{f_{0}-3\sigma }^{f_{0}+3\sigma }u_{y}\left(\xi ,\zeta ,2\pi f\right)E\left(f\right)df.$$
when d=1, x=7, a=1.5, and σ=0.03, by calculating the amplitude $M=|{\bar{u}}_{y}\left(\xi ,d/2,f\right)|$, the amplitude–frequency relationship in elastic and viscoelastic media in the frequency domain can be obtained [36]. For an elastic material, the amplitude is linearly attenuated with increasing transducer frequency. By contrast, the amplitude in a viscoelastic medium decays exponentially with an increasing transducer frequency and is nearly 0 at 3 MHz. Therefore, the excitation frequency of the transducer should be less than or equal to 1 MHz for a viscoelastic medium. According to the principle of ultrasonic testing, the lower the frequency of the transducer is, the longer the wavelength and the lower the resolution [37]. When the transducer frequency is 500 kHz and its wavelength is 8.7 mm, it is difficult to distinguish the smallest defect (artificial hole of ⌀2 mm × 10 mm). Therefore, 1 MHz was selected as the transducer frequency in the present work, and other frequencies are not discussed.

In many solutions of the transcendence equation, since 1 MHz was used as the frequency of the developed transducer, the frequency range of the independent variable was set from 950 to 1050 kHz. Solving the transcendental equation $kaJ_{0}\left(ka\right)=\left(1-\sigma \right)J_{1}\left(ka\right)$ through the graphical method can yield the resonance frequency of the four-laminated transducer, as shown in Figure 4.

Figure 4 shows that the resonance frequency of the transducer formed by stacking four 4 MHz piezoelectric wafers was about 1 MHz, which was consistent with the actual measured value of 1.03 MHz, with a relative error of 0.3%.

## 3. Simulation of Manufacturing and Experiments

### 3.1. Piezoelectric Materials

Piezoelectric materials play an important role in the sensitivity and flaw detection resolution of piezoelectric transducers. For the selection of piezoelectric materials, three principles should be followed [37]. First, the performance of the piezoelectric materials should be appropriate to meet the specific use requirements, and a high performance should not be excessively pursued. Second, the working performances of the piezoelectric materials should be stable and reliable. Finally, piezoelectric materials are cheap and easy to process. A sufficiently large electromechanical coupling coefficient $K_{t}$ is required to obtain a good conversion efficiency. A sufficiently small mechanical quality factor $\theta_{m}$ is required to obtain a good resolution. Relatively large values for the piezoelectric strain constant $d_{33}$ and the piezoelectric voltage constant $g_{33}$ are required to obtain a higher transmission sensitivity and reception sensitivity. A high Curie temperature $T_{c}$ and an appropriate acoustic impedance $Z_{p}=\rho \times c$, where ρ is the density of the piezoelectric material and *c* is the longitudinal wave velocity, are required. The key parameters of commonly used piezoelectric materials are listed in Table 3 based on a previously published report [37]. One of the leading zirconate titanate piezoelectric materials (PZT-5) provided the best comprehensive performance.

### 3.2. Simulation of Sound Field

The simulation model of the four-laminated transducer is shown in Figure 5, which was composed of a solid rocket propellant, piezoelectric wafers, and a copper sheet. The solid rocket propellant exhibited a 90° fan-shaped distribution with a radius of 36 mm. The radius, thickness, and frequency of the piezoelectric wafer were 12 mm, 0.5 mm, and 4 MHz, respectively. The radius of the copper sheet was the same as that of the piezoelectric wafer, and the thickness was 0.2 mm. The material parameters in the simulation process, such as the density, longitudinal wave velocity, and Poisson’s ratio, are shown in Table 4. Since the laminated transducer is an axisymmetric structure, its simulation model can be simplified to a quarter of the overall structure. In this way, not only is the accuracy of the simulation guaranteed but the calculation time can also be greatly reduced.

In the simulation process, the finite element method (FEM) in the COMSOL Multiphysics simulation software was used to simulate the sound field of the piezoelectric wafer through solid mechanics and electrostatics. A symmetric boundary condition was applied in the x-direction, the polarization direction between adjacent piezoelectric wafers was opposite, the upper surface of the piezoelectric wafer was loaded with Volt = 1, and the lower surface was grounded. The bottom surface of the piezoelectric wafer stack was set as a roller boundary condition to prevent the wafers from moving vertically, which was equivalent to the backing material in the transducer. The outer surface of the solid propellant was set as the radiation boundary condition of a spherical wave. On the one hand, it was used to realize the assumption that the boundary of the solid propellant was infinite, and on the other hand, spherical waves produce little reflection when they radiate outward from the geometric boundary. Additionally, the outer surface of the solid propellant was also set as a far-field calculation boundary, which helped to characterize the sound pressure distribution in the far-field limit.

The size of each element was set to one tenth of the wavelength to balance the calculation accuracy and speed. The analysis type was harmonic, the solution method was full, the frequency was 1 MHz, the step was 10 Hz, and the damping constant coefficient was 0.05. After the solution was completed, the directive curve of the piezoelectric transducer could be obtained by post-processing. Figure 6a,b show the results for a single-wafer transducer and the four-laminated transducer.

The half-angle spread refers to the intersection angle between the tangent of the main lobe and the adjacent side lobe and the axis of the main sound beam, also known as the pointing angle. Denoted by $\theta_{0}$, it is expressed as follows [37]:(31)$$\theta_{0}=\arcsin\left(1.22\times \frac{\lambda }{D_{s}}\right),$$
where λ is the wavelength and $D_{s}$ is the wafer size. In the simulation process, the sound velocity of PZT-5 was 4350 m/s and the wafer size was 24 mm. According to the calculation, the half-angle spreads of the single-wafer and four-laminated transducers were 12° and 3°, respectively. This was almost consistent with the simulation results. The smaller the half-angle spread is, the more concentrated the sound field energy is, and the higher the detection sensitivity becomes. Therefore, the four-laminated transducer had a higher sensitivity than the single-wafer transducer.

The energy distribution of the ultrasonic transducer was mainly concentrated on the main lobe and side lobe of sound field. As shown in Figure 6, for a single-wafer transducer, the gain of the main lobe was 170 dB, the maximum gain of the side lobe was 155 dB, and their distribution range was 120°. For the four-laminated transducer, the main lobe gain was 195 dB, the maximum gain of the side lobe was 165 dB, and their distribution range was also 120°. Therefore, under the condition of almost the same energy distribution range, the gain of the four-laminated transducer was 25 dB higher than that of the single-wafer transducer. The reason was that the laminated transducer used four 4 MHz piezoelectric wafers to form a 1-MHz cylindrical sound field through serial superposition. This further proved that the four-laminated ultrasonic transducer with a frequency of 1 MHz was feasible.

### 3.3. Piezoelectric Elements and Backing Material Bonding

Typically, the piezoelectric elements are directly bonded to the backing material by an adhesive. Therefore, the thickness of the adhesive layer must be as uniform and thin as possible with no bubbles. These requirements were met in the present work by employing an epoxy resin with a ratio of 1:10 as the adhesive during the transducer manufacturing process. After bonding, a specially designed holder was used to maintain an adequate contact pressure between the piezoelectric elements and the backing material during the curing process. Finally, the bonded piezoelectric elements and backing material, along with the holder, were placed together in an oven at 100 °C to cure for 4 h.

### 3.4. Electrode Wire Connections

Electrically conductive wires must be soldered carefully to the piezoelectric elements at an appropriate soldering temperature and for an appropriate soldering time to ensure optimal electrical connections, which are essential for maintaining good transducer performance. The present work used a silver-plated aviation wire with a 1.0 mm diameter for signal transmission. The solder joints were located directly in the center of the piezoelectric elements and were made as small as possible to reduce their impact on the ultrasonic transmission and reception fields. The lengths of the wires were generally 5–6 mm to facilitate subsequent soldering and performance testing.

### 3.5. Epoxy Potting

Under the ideal acoustic impedance matching condition, the acoustic impedance of the potting material should generally be two-thirds that of the piezoelectric elements. The values of the acoustic impedance $Z_{p}$ of optimal piezoelectric elements composed of PZT-5 are listed in Table 3. Therefore, we used a potting material composed of an epoxy and tungsten powder composite, where tungsten powder was used as a filler material within an epoxy resin. Here, the cured epoxy resin matrix provided the Young’s modulus of the potting material and also determined the velocity of the acoustic waves within the material. The tungsten powder filler material increased the density of the material, while also scattering and absorbing acoustic waves.

In the production process, the epoxy resin and tungsten powder were configured in a ratio of 1:6 and were heated after mixing with a hair dryer to remove bubbles. In addition, a satisfactory attenuation effect was achieved for a potting material thickness greater than the wavelength of the propagating acoustic waves by a factor of 10. Here, the velocity of the acoustic waves propagating in the mixed tungsten epoxy powder was about 1700 m/s at f=1 MHz. Therefore, the wavelength was 1.7 mm, and the thickness of the potting material should exceed 17 mm. In the actual production process, the thickness of the potting material was generally about 20 mm.

### 3.6. Experiment of Signal Waveform

The signal waveforms of different transducers are shown in Figure 7a,b for the single-wafer transducer and four-laminated transducer. The single-wafer transducer was V102-RB (V refers to a videoscan transducer, which is a transuduer that is mainly used to detect solid media with strong attenuation or easy scattering, 102 indicates that the probe test result is the transverse sound beam sound field distribution, and RB is the right angle of the BNC connector), which was manufactured by Panametrics (OLYMBUS, Waltham, MA, USA). Its frequency was 1 MHz, and its wafer size was 25 mm. The four-laminated transducer was developed by the authors. Its frequency was 1 MHz, and wafer size was 24 mm. During the experiment, the thickness of the solid rocket propellant was 36 mm, and the penetration method was used to obtain a higher amplitude.

For the solid rocket propellant of the same thickness, the gain of the single-wafer transducer was 35.6 dB, and that of the laminated transducer was 20.4 dB. That is, the gain of the laminated transducer was 15.2 dB higher than that of the single-wafer transducer. The noise of the single-wafer transducer was significantly higher than that of the four-laminated transducer in both amplitude and quantity, which was related to the gain between them. The gain was lower than that of the simulation result. The main reason for this was that the simulation was carried out under relatively ideal conditions.

## 4. Testing Equipment

### 4.1. Dual-Probe Pulse Echo Method

The structure of the dual-probe pulse echo method is illustrated in Figure 8. The solid rocket propellant was a cylinder with a central bore hole that emitted excitations from the transmitting transducer reflected from the central hole that were received by the receiving transducer. The two transducers were placed on the surface of the solid propellant and separated by an arc length that depended on the radius of the solid propellant. However, any defects, such as inclusions or holes, within the propagation path would also reflect some portion of the emitted acoustic wave, which would then be received by the receiving transducer in the form of a defect echo, while the remaining excitation would continue propagating until being received by the receiving transducer as a bottom echo. The amplitude of the defect echo was related to the position and cross-sectional area of the defect along the path of propagation. Accordingly, defects can be identified qualitatively and quantitatively according to the amplitude of the defect echo or bottom echo.

### 4.2. Testing Equipment

As shown in Figure 9, the testing equipment was composed primarily of a rotary encoder, a water pipe, a detection cantilever, transmitting and receiving transducers, a solid propellant, a mechanical transmission link, a water tank, a transmitter/receiver unit, an industrial computer, and a programmable logic controller (PLC) unit.

In the testing process, first, the solid propellant was manually placed on the four wheels of the transmission link, and the transmitting and receiving transducers with the cantilever were manually pressed smoothly against the surface of the solid propellant. The transducer frame was fitted with a spring to ensure that the transducers maintained good physical contact with the solid propellant during the scanning process. Second, the system software sent a start instruction to the PLC unit through a RS232 serial port. The PLC unit turned on the water pump to flush the interface between the solid propellant and the ultrasonic transducers with water, and turned on the rotary motor to rotate the propellant while simultaneously turning on the linear transfer motor to drive the detection cantilever forward and backward along the propellant axis to scan the solid propellant. The propellant rotation was then recorded by the rotary encoder to position the propellant while also serving as a trigger impulse to start the data acquisition card and transmitter/receiver unit. Finally, the transmitter/receiver unit sent an electrical signal to the transmitting transducer to generate ultrasonic waves, while the receiving transducer acquired the ultrasonic echo signals and transmitted them to the industrial computer through the transmitter/receiver unit and data acquisition card. The system software used signal processing algorithms to visually represent the flaws of the solid propellant. After the completion of the scanning over the given length, the software system issued a stop instruction to the PLC unit, which then shut off the water pump, rotary motor, and linear transfer motor. The details of the industrial control components are shown in Table 5, including the name, brand, and model.

## 5. Results and Analysis

### 5.1. Features of Artificial Defects

According to the principle of ultrasonic testing, the defect size was mainly determined by the signal amplitude. However, differences arose between the equivalent size of the defect and its true size because the amplitude of the defect signal depended not only on the size of the defect but also on its shape, location, and type [37]. Therefore, near-surface and internal hole defects were manufactured to evaluate the qualitative and quantitative detection for the solid rocket propellant, as shown in Figure 10. The lengths of near-surface defect holes were 10 mm, and their diameters were 2, 4, 6, 8, and 10 mm, respectively. The distance between them was 30 mm, and the angle between the axis of the cylindrical propellant and the near-surface holes was 45°. The single internal defect size was 10 mm, and its diameter was 10 mm. It was parallel to the propellant axis. In the production process, the solid rocket propellant was extruded along the axial direction, and its internal defects were distributed along the axial direction due to the axial stretching. To be consistent with the distribution direction of the natural defects, the fabrication of artificial holes generally presented an angle of 45${}^{{}^{\circ}}$ with the axial direction.

The echo signals obtained by the developed equipment are presented in Figure 11. As expected, the echo without a defect is shown in Figure 11a, which included only a bottom echo with the amplitude in the range of −145 to 130 mV. The echo of the near-surface defect is shown in Figure 11b, which indicated that the defect echo appeared approximately 60 μs prior to the bottom echo, and the amplitude of the bottom echo was relatively small, with a range of −120 to 95 mV. The main reason was that some incident acoustic waves were reflected by the defect rather than by the central hole of the propellant. That is, the position of the surface defect was far from the central hole, and the direction of the hole was at an angle of 45° relative to the direction of the incident acoustic wave. The echo of the internal defect is shown in Figure 11c, which indicated not only that the defect echo appeared between the bottom echos but also that the amplitude of the bottom echo was substantially reduced, with a range of −98 to 75 mV. In this case, the defect echo and the bottom echo were detected simultaneously with no time difference, and their amplitudes were quite similar. That is, the internal defect was positioned very close to the central hole of the propellant, and the direction of the defect was perpendicular to the direction of the incident acoustic wave.

### 5.2. Artificial Defects

The detection results obtained for the artificial hole defects are shown in Figure 12. Here, the x-axis represents the length of the solid propellant, ranging from 0 to 1500 mm, and the y-axis is the rotational angle of the sample, ranging from 0° to 360°. The defects in Figure 12 marked 1–5 represent the near-surface defects, and the marked 6 represents the internal defect. The defect image area basically increased linearly for the near-surface defects with an increasing hole diameter. The dual-probe mode provided a defect echo on the path of the acoustic wave propagation for both the transmitting transducer and the receiving transducer. Therefore, each artificial hole defect resulted in two near-surface defect images that were symmetrically distributed above and below the central axis of the propellant. In contrast, the image of the internal defect was no longer symmetrically distributed with respect to the central axis of the propellant but appeared as a somewhat uniform and continuous defect image, which is the result of the combined effect of its orientation and size.

### 5.3. Natural Defects

For two different natural defects, the results of the ultrasonic method are shown in Figure 13a,c, and those obtained by industrial CT imaging are shown in Figure 13b,d. First, from the perspective of the defect shape, for natural defect I, the detection results of the two methods were quite different, while for natural defect II, the detection results of the two methods were almost the same. Second, from the defect area, for natural defect I, the defect area of ultrasonic imaging was significantly larger than that of the industrial CT, while for natural defect II, the two areas were almost the same. Finally, from the angle range of the defect distribution, for natural defect I, the ultrasonic imaging angle ranged from 280° to 330°, and the angle range of the industrial CT was about 20°. There was a significant difference between the two. For natural defect II, the angle range of the ultrasound imaging was from 280° to 350°, and the angle range of industrial CT was about 30°. There was also a significant difference between the two. The analysis of the detection results showed that although the two detection methods could accurately detect the two natural defects, from the perspective of quantitative detection, the ultrasonic method was more accurate in the characterization of the two natural defects. The main reason was that the industrial CT could not detect the defects parallel to the ray’s incident direction. The results further verified that the four-laminated transducer with a frequency of 1 MHz obtained a better solution for evaluating the security of the solid rocket propellant.

## 6. Conclusions

Based on the principle of underwater transducers, the following tasks were completed in this study:Based on the correspondence principle of simple harmonic waves, the wave equation in viscoelastic media was derived. Since the equation is a function of the angular frequency ω, an ultrasonic wave propagates in a viscoelastic medium with greater attenuation than that in an elastic medium.According to the amplitude–frequency relationship in viscoelastic media, by setting the independent variable range of 950–1050 kHz for the transcendental equation, the 1-MHz resonance frequency of the laminated transducer was obtained by a graphical method.The sound field simulation and experimental results showed that the gain of the four-stack transducer was 15 dB higher than that of the single-element transducer.An ultrasonic feature scanning system was built to realize the qualitative and quantitative detection of the smallest artificial hole (⌀2 mm × 10 mm).Two different natural defects were scanned and the results were compared with those obtained using an industrial computed tomography detection system. The results showed that the ultrasonic method was more accurate in the characterization of two natural defects. The primary cause was that the industrial CT was not sensitive to defects parallel to the incident direction of the ray.

The research presented in the paper not only provides a method for the qualitative and quantitative non-destructive testing of solid propellants, but also provides an important reference for other viscoelastic components.

## Figures and Tables

**Figure 1 sensors-21-07188-f001:**
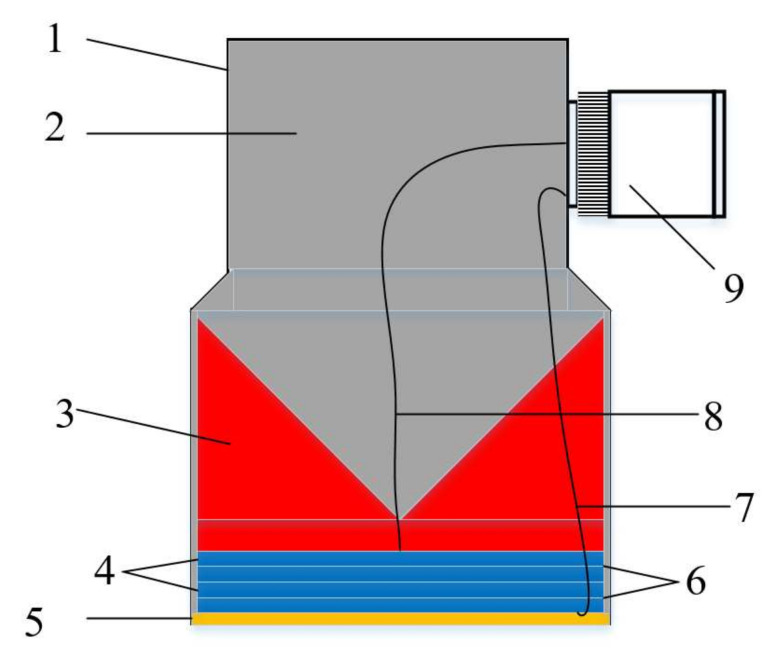
Structure of four-laminated transducer (1. case, 2. epoxy potting, 3. backing material, 4. piezoelectric wafers, 5. matching layer, 6. copper sheet, 7. ground wire, 8. signal wire, and 9. connector).

**Figure 2 sensors-21-07188-f002:**
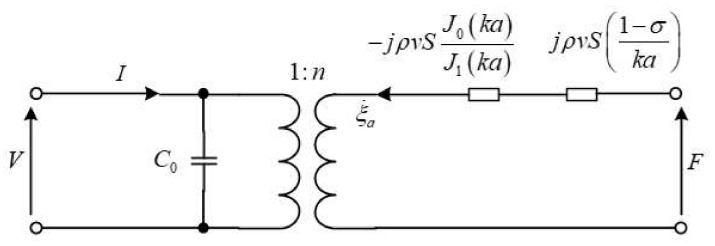
Electromechanical equivalent diagram of single wafer.

**Figure 3 sensors-21-07188-f003:**
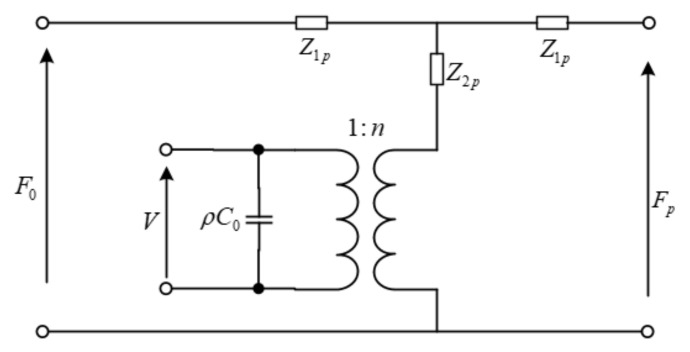
Electromechanical equivalent diagram of the laminated transducer.

**Figure 4 sensors-21-07188-f004:**
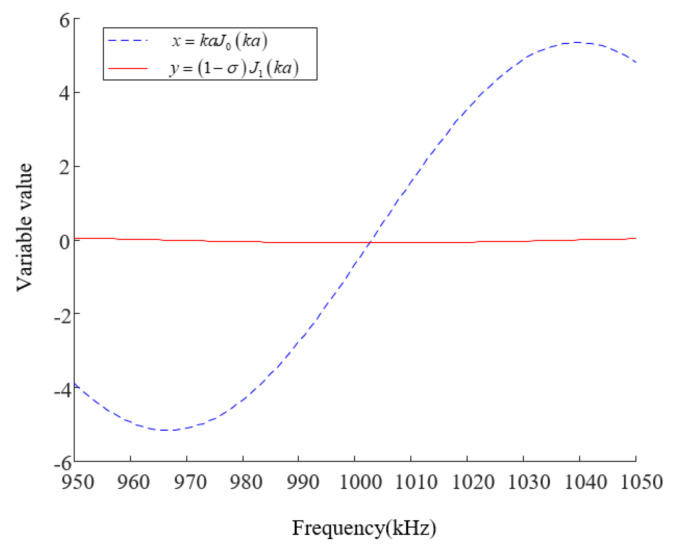
Calculated curve of the transcendental equation (a=12 mm, $s_{11}^{E}=12.3\times 10^{-12}$ m${}^{2}$/N, $s_{12}^{E}=-4.05\times 10^{-12}$ m${}^{2}$/N, ρ=7500 kg/m${}^{3}$).

**Figure 5 sensors-21-07188-f005:**
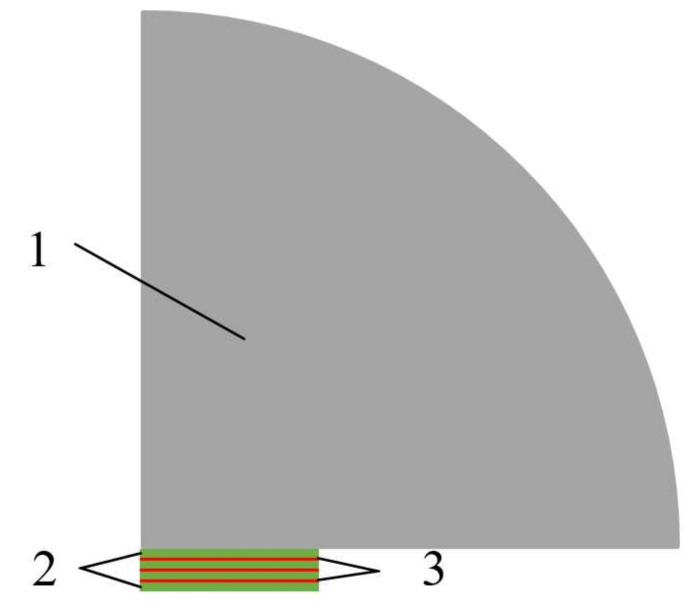
Simulation model of the four-laminated transducer (1. solid rocket propellants, 2. piezoelectric wafers, and 3. copper plate).

**Figure 6 sensors-21-07188-f006:**
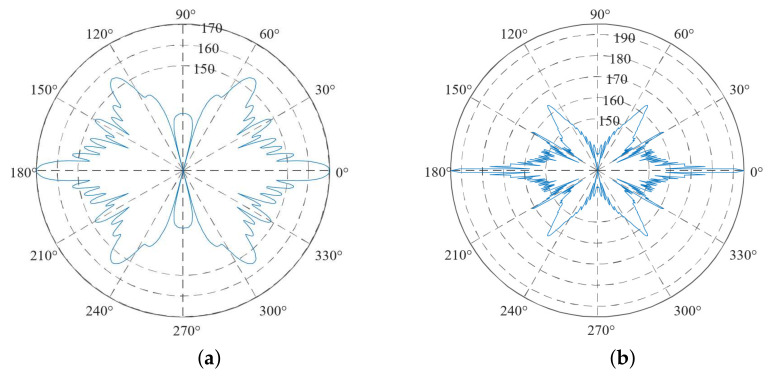
Directive curve of ultrasonic transducer ((**a**) single-wafer transducer and (**b**) four-laminated transducer).

**Figure 7 sensors-21-07188-f007:**
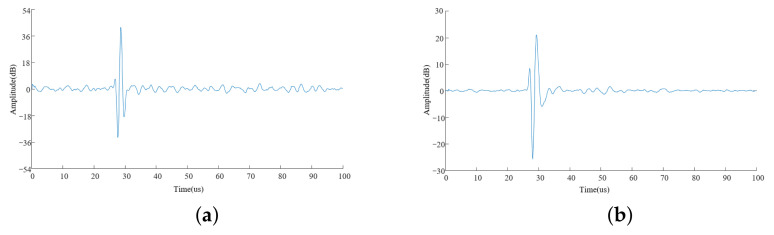
Experiment results ((**a**) V102-RB (35.6 dB) and (**b**) developed transducer (20.4 dB)).

**Figure 8 sensors-21-07188-f008:**
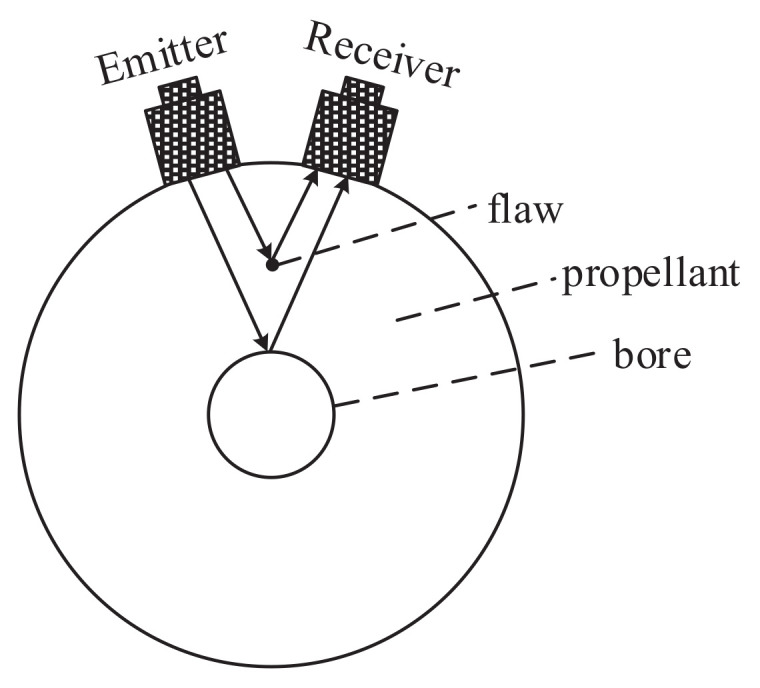
Schematic diagram of the dual-probe echo method.

**Figure 9 sensors-21-07188-f009:**
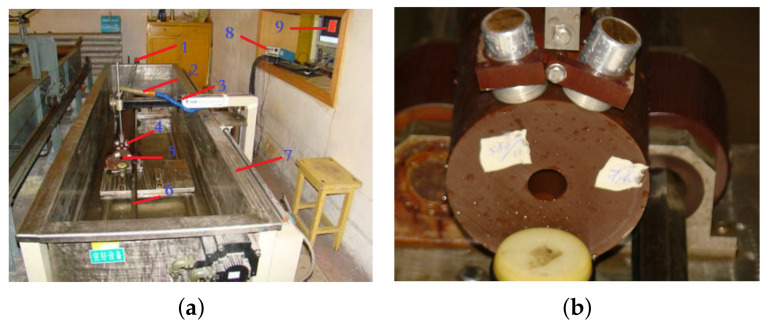
Appearance of testing equipment ((**a**) testing equipment: 1. rotary encoder, 2. water pipe, 3. detection cantilever, 4. transmitting and receiving transducers, 5. solid propellant, 6. mechanical transmission link, 7. water tank, 8. transmitter/receiver unit, and 9. industrial computer), (**b**) transducer and propellant).

**Figure 10 sensors-21-07188-f010:**
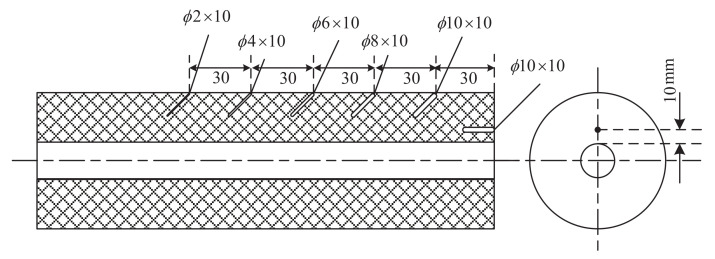
Sizes and distribution of artificial defects (the holes are defined as diameter × depth with all units in mm).

**Figure 11 sensors-21-07188-f011:**
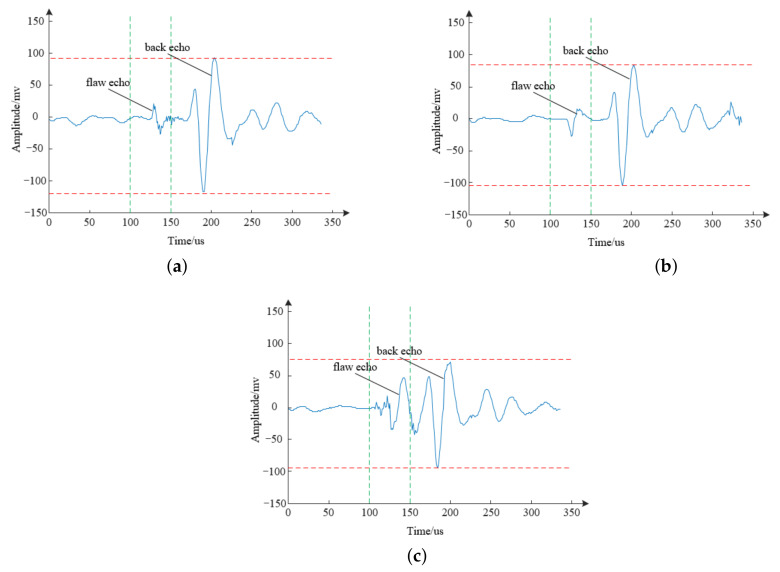
Different ultrasonic echo ((**a**) defect-free echo, (**b**) near-surface defect echo, and (**c**) internal defect echo).

**Figure 12 sensors-21-07188-f012:**
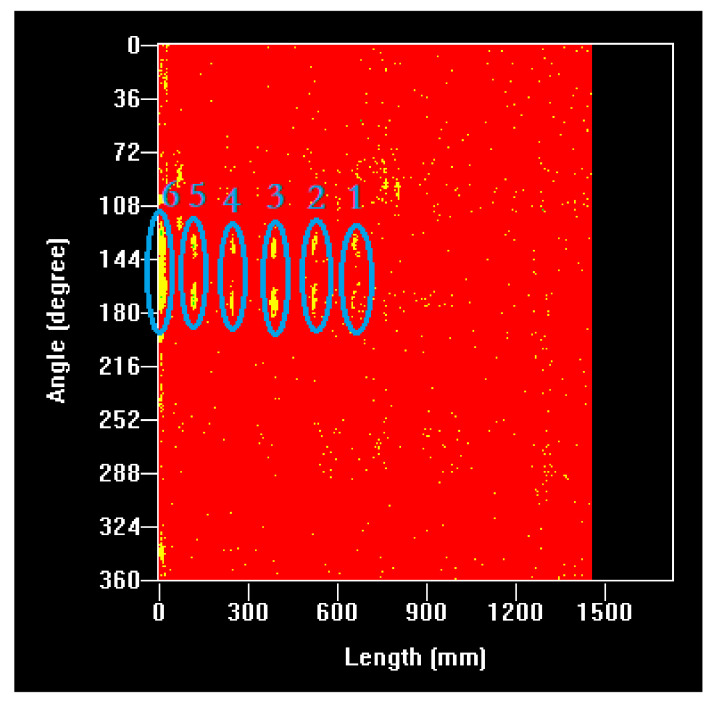
Artificial defects (near-surface defects: 1. ⌀2×10, 2. ⌀4×10, 3. ⌀6×10, 4. ⌀8×10, and 5. ⌀10×10, internal defect: 6. ⌀10×10).

**Figure 13 sensors-21-07188-f013:**
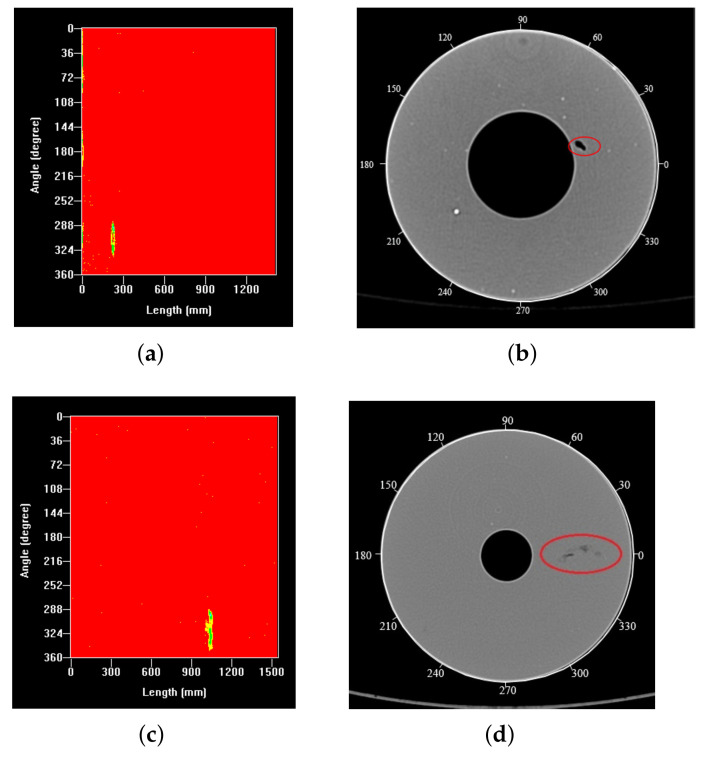
Natural defects ((**a**) natural defect I(UT), (**b**) natural defect I(CT), (**c**) natural defect II(UT), and (**d**) natural defect II(CT)).

**Table 1 sensors-21-07188-t001:** Differences between proposed and underwater transducer.

Name	Proposed Transducer	Underwater Transducer
On wafers	Backing material	Rubber sheet
Laminated wafers	Solid wafer	Hollow wafer/Curved wafer
Under wafers	Matching layer	Rubber sheet
Application	Ultrasonic testing	Communication/Target detection

**Table 2 sensors-21-07188-t002:** Correspondence between elastic and viscoelastic media.

Name	Elastic Media	Viscoelastic Media
Shear modulus	μ	$\mu^{*}\left(i\omega \right)$
Lamé constant	λ	$\lambda^{*}\left(i\omega \right)$
Bulk modulus	*K*	$K^{*}\left(i\omega \right)$
Modulus of elasticity	*E*	$E^{*}\left(i\omega \right)$
Poisson’s ratio	υ	$\upsilon^{*}\left(i\omega \right)$

**Table 3 sensors-21-07188-t003:** Key parameters of piezoelectric materials.

Material	$d_{33}^{1}$	$g_{33}^{2}$	$K_{t}$	$c^{3}$	$Z^{4}$	$\theta_{m}$	$T_{c}^{5}$
Quartz	2.31	5.00	0.10	5740	15.2	$10^{4}$	550
Lithium sulfate	16.00	17.50	0.30	5470	11.2	$10^{4}$	75
Lithium iodate	18.10	32.00	0.51	4130	18.5	<100	256
Barium niobate	6.00	2.30	0.49	7400	34.8	>$10^{5}$	1200
Braium titanate	190.00	1.80	0.38	5470	30.0	300	115
Lead titanate	58.00	3.30	0.43	4240	32.8	1050	460
PZT-4	289.00	2.60	0.51	4000	30.0	500	328
PZT-5	374.00	2.48	0.49	4350	33.7	75	365
PZT-8	2.50	2.50	0.48	4350	33.0	1000	300

$d_{33}$: piezoelectric strain constant, $d_{33}\times 10^{-12}$ (m/V). $g_{33}$: piezoelectric voltage constant, $g_{33}\times 10^{-3}$ (V·m/N). *c*: velocity of longitudinal wave, *c* (m/s). *Z*: acoustic impedance, $Z\times 10^{5}$ (g/cm${}^{2}\cdot$s). $T_{c}:T_{c}$ (°C).

**Table 4 sensors-21-07188-t004:** Parameters of key components.

Material	Density (kg/m${}^{3}$)	Velocity (m/s)	Poisson’s Ratio [37]
Solid rocket propellant	1500	2000	0.5
Piezoelectric crystal plate (PZT-5)	7750	4350	0.34
Copper plate	8600	4700	0.37

**Table 5 sensors-21-07188-t005:** Details of the industrial control components.

Name	Brand	Model
Industrial computer	Advantech	IPC-610-H
PLC	MITSUBISHI	FX2N-64MR-001
Squarewave pulser/receiver	PANAMETRICS	5077PR
Servo motor	YASKAWA	SGM7D
Rotary encoder	AUTONIC	E40S6

## Data Availability

The data presented in this study are available on request from the corresponding author. The data are not publicly available because we are creating an industrial data set.

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
