# Peer review of "An Ultrasonic Laminated Transducer for Viscoelastic Media Detection"

_sensors, 2021, doi:10.3390/s21217188_

Round 1

Reviewer 1 Report

The manuscript is overall well written (with some corrections to be made), well detailed and would deserve to be published after modifications and corrections.

The symbol "*" is a temporal convolution product, the authors must specify it.
The equations (1-4) are written in the time domain with a time dependence of lambda(t) and mu(t), however in the equations (6-8) we do not see this time dependence anymore, what happened? did we pass from the temporal to the frequency domain without specifying it and keeping the same notations? It is necessary to clarify this aspect which does not give a coherent vision of the equations. Are lamba* and mu* the Fourrier transforms of lambda and mu, since at the beginning of the equations lambda and mu were two functions of time?

Author Response

Dear Reviewer:

        Please see the attachment, thanks a lot!

Reviewer 2 Report

The article reports on an interesting investigation.  
Nevertheless, it would be advantageous for the article to revise some points:
-English should be revised
-the structure of the document does not correspond to that of a classic article (introduction divided into different sub-parts, experimental part too small, etc.)
-acronyms and variables are not defined (eg TVR, µ*, etc.)
-some legends are not consistent with the associated figures (eg Figure 4)
- units are misused either in the text or in the figure (eg Figure 7:amplitude in db to be consistent with the text)
- figures are little or badly used (Fig 4: specify resonance and anti-resonance, use a wider frequency range and discuss other modes, Figure 6: extend analysis, Figure 8: add a zoom on the transducer produced and on the tested sample)
- details should be given regarding the technical means (which method used for the simulation? which means? which hypotheses? which parameters? ...)
- Table 2 is restricted to its simplest expression regarding the parameters of the piezoelectric material, why the choice of a PZT 5H? is this consistent with the final application? what are the materials used for the experimental transducer? has it been purchased?
- details on obtaining figure 11 should be given and units should be added to figure
- details of the industrial technical control must be provided (brand, model, etc.)
- finally the authors should give more details about their experiments (e.g. why 45° holes?) and go further in the analysis (e.g. For Fig. 12 (c), and (d), the angle of the defect is 75°, and its length is about 45 mm. In this case, the result of ultrasonic method is completely consistent with that obtained by industrial CT method)

Author Response

(The authors gave the same response as above.)

Reviewer 3 Report

(1) the background information was too much introduced in contrast to the results and analysis part. actually readers are more interestged in the  resutls analysis part which can be extended if possible.  

(2) for the first two conclusions, in addition to the general pattern, the authors are expected to insert more technical contents. 

(3) some expressions need to be corrected to be better understood, such as the title of table "main material parameters", what does "main" represent? Additionally, please specify how the poisson's ration was obtained, from laboratory test or from reference? 

Author Response

(The authors gave the same response as above.)

Round 2

Reviewer 1 Report

I find this form of the article much better, the authors have made a real effort to improve the presentation and interoperation of the results. I find this form of the article much better, the authors have made a real effort to improve the presentation and interoperation of the results.